# One Shell of a Problem: Cumulative Threat Analysis of Male Sea Turtles Indicates High Anthropogenic Threat for Migratory Individuals and Gulf of Mexico Residents

Micah Ashford [1], James I. Watling [1] and Kristen Hart [2,*]

[1] Department of Biology, John Carroll University, University Heights, OH 44118, USA
[2] U.S. Geological Survey, Reston, VA 20192, USA
* Correspondence: kristen_hart@usgs.gov; Tel.: +1-011-954-650-0336

**Abstract:** Human use of oceans has dramatically increased in the 21st century. Sea turtles are vulnerable to anthropogenic stressors in the marine environment because of lengthy migrations between foraging and breeding sites, often along coastal migration corridors. Little is known about how movement and threat interact specifically for male sea turtles. To better understand male sea turtle movement and the threats they encounter, we satellite-tagged 40 adult male sea turtles of four different species. We calculated movement patterns using state-space modeling (SSM), and quantified threats in seven unique categories; shipping, fishing, light pollution, oil rigs, proximity to coast, marine protected area (MPA) status, and location within or outside of the U.S. Exclusive Economic Zone (EEZ). We found significantly higher threat severity in northern and southern latitudes for green turtles (*Chelonia mydas*) and Kemp's ridleys (*Lepidochelys kempii*) in our study area. Those threats were pervasive, with only 35.9% of SSM points encountering no high threat exposure, of which 47% belong to just two individuals. Kemp's ridleys were most exposed to high threats among tested species. Lastly, turtles within MPA boundaries face significantly lower threat exposure, indicating MPAs could be a useful conservation tool.

**Keywords:** sea turtle; cumulative threat; conservation

## 1. Introduction

Human use of oceans has dramatically increased in the 21st century. Results from a five-year study found that by 2013, 66% of the world's oceans faced increased pressure from anthropogenic activities like fishing, pollution, global shipping, and elevated sea surface temperatures [1]. Some of the highest cumulative impact scores (top 5% of scores) were in the Gulf of Mexico, Caribbean, and Central Atlantic Ocean [1]. Additional studies on human impact have found that the Caribbean scores among the highest regions in the world for marine-associated threats [2]. Due to high pressure from human activity, species within the Caribbean show the greatest rates of loss within neritic (coastal) habitats, where the highest species richness occurs [3]. In the Gulf of Mexico, more than 79 species, which represent 7 of 9 marine reptiles, 5 of 27 marine mammals, and 19 of 44 shark species, are listed as at least Near Threatened by the International Union for Conservation of Nature (IUCN) [4]. Additional research has also found that more than 26% of endemic bony fish in the Gulf of Mexico and Caribbean are facing severe population declines [3]. High cumulative human impact on marine wildlife is disruptive and can substantially increase the chance of species extirpation or extinction [5].

One of the major ways that human activities affect marine animals is by disrupting foraging and breeding behavior. Such disruptions force animals out of optimal habitats, which can decrease an individual's overall fitness [6–10]. Furthermore, marine wildlife that remain near human activity often experience reduced foraging time with diminished catch because they spend time avoiding interaction with humans due to stressors such as noise

from ship traffic [11–13]. Habituation to human activity can also be dangerous because it increases the risk of injury or death from boat strikes, capture, or aggressive behavior from conspecifics [14–16]. Habituation has also been documented to decrease health and increase disease risk in marine species due to improper diet [17].

Wildlife can also be disrupted by the indirect effects of human activity. Studies have shown that threats can occur along spatial gradients, as is the case with marine debris from areas with heavy shipping and fishing presence [18]. Similarly, chemical disturbances, such as those from polychlorinated biphenyls (PCBs) and oil spills, are often most concentrated at the source of contamination, but with wider ranging, although more diffuse effects through bioaccumulation or dispersal of affected individuals [19,20]. Additional pressures from threats like fishing and tourism can also be spatially autocorrelated, with high areas of pressure found closer to the shore or in clusters where activity is greatest [16,21]. However, of the disruptive activities on marine wildlife, fishing is regarded as highly destructive and can lead to irreversible effects on fisheries [22].

Fishing practices in the 21st century have exhausted multiple fisheries across the globe, to the point that only 32% have stocks above replenishable levels [22]. Current data on fisheries indicate that even as the size of the global fishing fleet has more than doubled from 1950 to 2017, catch per unit of effort of fish during that period decreased by more than 80% in most areas—evidence that global fish abundance may be rapidly declining [23,24]. Additionally, the world's fishing fleet uses methods that catch many species in non-target trophic groups, which is referred to as bycatch [25–27]. Despite upgrades in fishing technology to reduce bycatch, oversight to ensure that newer technology is being used is lacking [26]. Fishing methods meant to catch a target species are often indiscriminate [23,26]. Undesirable fish are discarded at sea, and due to lack of a governing body or trained individuals to observe active fishing vessels, especially on artisanal fishing boats, the catch goes under-reported, or unreported [26,28].

The pressures caused by vessels from the world's fishing fleet put fish and other marine species at risk, either directly from bycatch, or indirectly from boat strikes and pollution. Species that conduct long migrations are particularly at risk, such as elasmobranchs [29,30], pinnipeds [31,32], cetaceans [33,34], and chelonids [27,35–37]. To best protect migratory marine species, researchers rely on tracking individuals to better understand the threats they encounter within and en-route to their feeding and breeding locations [7,29–31,35,36,38–44]. Locations where species of interest aggregate can be used by managers to design strategies that protect them, for example by establishing or expanding marine protected areas (MPAs) [7,39,42]. If placed correctly, MPAs can be beneficial at protecting migratory species in addition to resident species of the locale, particularly those that hold economic value like species consumed by humans [45–48].

As they represent a well-known, charismatic group of migratory marine megafauna, sea turtles have been used to justify the establishment of MPAs. For example, Mexico [49], Gabon [50,51], and Indonesia [52] have established MPAs in areas where tracking studies found high use by sea turtles [53]. Sea turtle tracking studies have also helped managers create zones that prohibit oil exploration or pipe laying in areas that intersect migration routes [53,54]. Sea turtles also benefit from previously established MPAs. Large sea turtle aggregations of multiple species can be found within U.S. MPAs, such as the Florida Keys National Marine Sanctuary and Dry Tortugas National Park [55].

Established in 1990 and 1992, respectively, to create critical marine habitat and protect marine resources for a number of imperiled species, the Florida Keys National Marine Sanctuary and Dry Tortugas National Park encompass over 2900 square nautical miles and contain a mix of restricted and prohibited human activity zones, also known as marine zoning [56–58]. The Florida Keys National Marine Sanctuary and the reserves it overlaps, including Dry Tortugas National Park, have been found to have positive impacts, such as increasing population numbers for marine species, including green and loggerhead turtles and sharks, and species of economic value such as spiny lobsters (Family Palinuri-

dae) [39,59–61]. Both marine reserves show high use from threatened and endangered sea turtles [39].

All seven, extant species of sea turtle face population pressures from anthropogenic influence [62–68]. The IUCN has assessed loggerhead (*Caretta caretta*) and olive ridley (*Lepidochelys olivacea*) sea turtles as Vulnerable, leatherback (*Dermochelys coriacea*) sea turtles as Threatened, green (*Chelonia mydas*) sea turtles as Endangered, and hawksbill (*Eretmochelys imbricata*) and Kemp's ridley (*Lepidochelys kempii*) sea turtles as Critically Endangered [62–67]. Australian flatback turtles (*Natator depressus*) are Data Deficient [68]. During nesting years, adult female sea turtles will undergo migrations of up to thousands of kilometers to their natal beaches to nest, often traveling through areas with high human activity [7,44,69–71]. Female turtles aggregate in the vicinity of nesting sites for the duration of the mating season, where they lay multiple clutches of eggs before returning to their foraging grounds [72,73]. Fidelity to nesting locations each season is high, often in neritic habitats, which puts sea turtles at risk of human–wildlife conflicts [7]. For example, small-scale fisheries in Greece report heavy interaction involving loggerhead turtle bycatch in nearshore waters annually, one of the primary nesting locations for Mediterranean loggerhead turtles [74].

In response to human presence, turtles may be moving to less ideal habitats where interactions are fewer. For example, in Zakynthos Island, Greece, Schofield et al. [8] determined that during the 2020 lockdown due to the COVID-19 pandemic, nesting female sea turtles moved to warmer waters closer to shore that were previously occupied by high densities of humans versus staying farther offshore in colder waters. Turtles moving to warmer waters when humans were absent indicates they were residing in lower quality habitats to avoid human interactions—a behavior not detected until disruption of the daily pressures posed by the tourist industry. The phenomenon of species changing their behaviors to avoid humans has been well documented. Multiple mammalian species across the globe have shifted to nocturnal foraging patterns in response to heavy human presence, with recreation, resource harvesting, extractive activities, development, and vehicles being some of the leading causes behind this shift [75]. Species will also avoid areas occupied by humans or other predators, even if the area is richer in resources [76]. In marine species, this has been less documented, but has been found in sperm whales (*Physeter macrosephalus*) [77] and killer whales (*Orcinus orca*) [78]. Green sea turtles have been found to accept trade-offs in risk behavior. Turtles in good health have been found to avoid areas where predation risk is high, whereas those in poor health will be more risk prone to forage [79]. The recent paper by Schofield et al. [8] also indicates that turtles may be exhibiting similar behaviors of avoiding areas with heavy human presence that warrant further study.

Although how sea turtles respond to human threat is understudied, strong evidence suggests that sea turtles face multiple anthropogenic threats throughout their range [36]. However, due to differences in habitat, species may vary in the degree to which they encounter various threats [40]. Kemp's ridleys, for example, nest almost exclusively in Rancho Nuevo, Mexico, and are primarily found in the Gulf of Mexico, exposing them to a larger number of threats derived from oil pollution than hawksbill sea turtles, who are more confined to coral reefs and other tropical climates, at least within Caribbean, Southern Gulf of Mexico, and Atlantic populations [20,44,80–82]. Another study found that female Kemp's ridleys have much larger foraging ranges than loggerhead turtles, and minimal overlap of foraging grounds within the Gulf of Mexico, exhibiting spatial partitioning of habitat, which could indicate different exposure to certain human threats [40]. Foraging and migration timing also differs by species, even in areas where population overlap occurs. One study found that although a foraging ground in Florida was shared by three species of sea turtle (loggerhead, green, and Kemp's ridley turtles), spatiotemporal partitioning existed in dive depth, duration, and the time at which diving behavior occurred [83]. Seasonal migration timing may also play a factor in threat exposure, as different populations of species will move out of shared spaces at different times. One study found that in a seagrass bed used by three species of sea turtle, Kemp's ridleys and loggerhead turtles left the area when water temperatures dropped, but green turtles remained through the winter



season [84]. Therefore, due to differences in behavior, foraging location, and migration patterns, anthropogenic threats may affect the various species in different ways.

Data regarding sea turtles, although extensive, is mostly garnered from studies of females, due to the relative ease of capture on beaches when they nest [85–87]. Male sea turtles, however, are understudied, because they spend their entire lives in the ocean where in-water captures are more logistically difficult and financially expensive [42,87,88]. To date, the largest sample size of male sea turtles has come from Schofield et al. [7], who were able to track and record 38 adult, male loggerhead turtles, of which only five were tracked for more than one season. Furthermore, Schofield et al. found that male mortality may differ from females, primarily due to male energy uses towards breeding [89]. As most studies have small sample sizes and cover short temporal scales, data in the literature regarding male sea turtles are deficient. This is especially true for studies of migration, residency areas during non-breeding times, and the anthropogenic threats they face in those locations.

The tracking of male sea turtles is important from a conservation standpoint, as they exhibit different movement patterns than females [42,44]. While some male sea turtles reside near nesting beaches year-round, others have been found to exhibit long-distance migrations between breeding and feeding grounds with unique timing compared to females [42,44,90]. Migrations often remove species from protected areas into locations that increase an individual's risk of mortality from human interaction, such as artisanal and benthic fishing fleets in international waters or boundaries of countries where protection is weak [35,91]. Additionally, shipping lanes, and pollution from ships and oil platforms increase mortality risk [35]. Light pollution can negatively affect adult female sea turtles in addition to hatchlings [92,93]. Thus far, however, the risks posed by these threats have not been systematically evaluated for male sea turtles.

Male sea turtles will be increasingly affected by human threats as climate change accelerates in the 21st century. When coupled with sea level rise and coastal development, nesting beach habitat and therefore recruitment in these already small populations will be further reduced [94–96]. Additionally, sea turtles are long lived, taking anywhere from 8 –24 years to reach sexual maturity [97–99]. Replacement of lost males in a population is therefore slow, and data on male mortality rates are nonexistent in most locations [88]. Migration routes and phenology may also differ in male sea turtles compared to females [7,42,44]. Lastly, operational sex ratios and the reproductive value of male sea turtles is unknown, as females are focal points of studies relating to reproduction [88]. As such, the behaviors and movements of male sea turtles could be studied and examined to better understand the threats they face for conservation management efforts.

The purpose of this study was to track four species of male sea turtles to better understand their exposure to spatially and temporally variable threats. We focus specifically on threats faced by males within foraging areas and during long migrations between breeding and feeding grounds by using seven unique threat categories: (1) within or outside of an MPA boundary; (2) within 10 km of a coastline; (3) within the Exclusive Economic Zone of the United States (U.S.) or not; (4) fishing; (5) shipping; (6) oil rig proximity; and (7) light pollution levels. We predicted that (1) tagged turtles in our dataset would conduct long-distance migrations that will put them into contact with threats; (2) threat intensity would vary along a spatial and temporal gradient with increasing distance from MPAs like the Florida Keys National Marine Sanctuary and Dry Tortugas National Park; (3) exposure to threats would vary by species; (4) threats would be lower within MPA boundaries than outside of them; and (5) areas of high turtle point concentration would have lower threat values than areas with lower turtle presence.

## 2. Materials and Methods

### 2.1. Study Area/Species Collection

We captured turtles as in Hart et al. [100] from 2009–2019. Forty adult male sea turtles of four different species were captured from four locations using boat (jumping from boat,

snorkeling) or net capture via trawler. Sample sizes are as follows: Kemp's ridley = 6, hawksbill = 1, loggerhead = 8, green = 25. Capture location sample sizes are as follows: Dry Tortugas National Park = 24, Florida Keys National Marine Sanctuary = 6, Northern Gulf of Mexico = 9, Buck Island National Reef Monument = 1. We followed standard morphometric data collection, and attached platform transmitter terminals (PTT) to each turtle carapace using slow-curing epoxy (two-part Superbond epoxy; see Hart et al. [100]). Turtles were tracked using Wildlife Computers (Redland, WA, USA) SPOT or SPLASH10 transmitters. Tracking data ranged from 8 June 2009 to 7 August 2020 [100,101].

## 2.2. Collection and Calculation of Threats/State-Space Modelling

We performed a switching state space model (SSM) on the raw location data in order to estimate each turtle's true location at regular time intervals due to significant positional uncertainty in the raw satellite data [101]. Briefly, we used a Bayesian hierarchical movement model implemented in the R package 'bsam', using the 'hDCRWS' model specification and a time step of 1 day [102–104]. We set the Markov Chain Monte Carlo (MCMC) parameters following Roberts et al. [105], which used adaptive sampling for 7000 draws, taking 10,000 samples from the posterior distribution, and then thinning by five to reduce MCMC autocorrelation, resulting in 2000 posterior samples from which to make inference. This process ultimately resulted in an improved dataset by eliminating location errors and provided one location point for each turtle per day.

We collected a total of 8875 SSM points for threat analysis [101]. Through review of scientific literature and professional consultation, we collected data for seven primary threats to male sea turtles (Fishing, Shipping, Drilling Platforms, Light Pollution, MPA boundaries, located within or outside the U.S. Exclusive Economic Zone (EEZ), and coastal threat (within 10 km of a coastline)) [8,35,36,101,106–109]. Raw location data have spatial accuracy ranges that vary between 500 m to 1.5 km. Raw tracking data were therefore fit to a hierarchical, behavior-switching state-space model (SSM), which was then used to increase the accuracy of tracking data and to determine home ranges of each individual [103]. This allowed for estimation of the behavioral modes of individual turtles (unique behavioral patterns), regularization of the locations in time, and accounting for location error in the raw satellite data. In order to accurately depict the threats within the area of each SSM point (1.5 km), we created a 2-km radius buffer around each SSM turtle point using R [104], within which threats were assessed. The threat data were collected and prepared as described in the following sections.

## 2.3. Fishing Data

Threats from fishing can come from a variety of sources (artisanal, longlining, commercial, nets and trawlers, etc.). Since 2016, all commercial fishing vessels within U.S. waters over 65 feet in length are required to have an AIS (Automatic Identification System) transponder tag attached, which tracks the ships every hour via satellite global positioning system (GPS) and ground-based receivers placed by the U.S. Coast Guard [110]. At present, only 2% of the world's fishing vessels have AIS tags on board, but these ships account for more than 50% of total fishing efforts [110]. We used a fishing density raster layer of ground-based, AIS-tracked ships as a representative subsample of fishing fleet intensity from marinecadastre.gov, a joint collaborative data repository for marine-related research by the National Oceanic Atmospheric Administration (NOAA), and the Bureau of Ocean Energy Management (BOEM) [111]. This layer includes tracks of fishing vessels that leave U.S. waters in the Gulf of Mexico, and therefore provides a representative subsample for turtles that move beyond the U.S. EEZ boundary.

Fishing intensity was measured in grid cells 1 hectare in size, with each cell representing the total number of fishing craft that passed through that cell with an AIS transponder onboard within a given year. We added and averaged the total number of fishing vessels per cell from 2015–2017 to get the mean fishing intensity per cell. We then created a single raster layer for analysis using ArcGIS Pro ver. 2.5 [112]. Although only three years of

fishing data were available, we assume fishing density was similar enough in previous years that the average cell values of the three years represent past fishing seasons. We averaged the fishing intensity score of all raster cells within each SSM turtle buffer and assigned that value as the fishing threat score for that point in ArcGIS Pro ver. 2.5 [112].

### 2.4. Shipping Data

Shipping data were also obtained using AIS tagged ships, and were downloaded in their raw format courtesy of the U.S. Coast Guard with certain identifiers scrubbed for privacy. We were able to obtain data for the entire study period from the Marine Cadastre data repository [111]. These data cover the entire study area and are a suitable, representative sub-sample of shipping data, clearly showing all shipping lanes within our study area.

AIS-tagged ships in the United States account for 50–60% of shipping activity [109]. We clipped all AIS data to each 2 km turtle buffer by date and then merged the data into a single vector shapefile to create a layer of shipping points that coincide with the presence of each 2 km SSM turtle buffer. Due to the large size of the AIS data files, we ran an RStudio instance on Google's Cloud Computing Engine [104]. The total number of shipping points within each 2 km buffer was then added and assigned as the threat score for that SSM point.

### 2.5. Drilling Platform Data

We downloaded drilling platforms point data, also referred to as oil rigs, oil platforms or drilling rigs to represent oil derived threats from the Marine Cadastre data repository [111]. We calculated the number of platforms within each turtle buffer by clipping the oil rig layer to the turtle layer and merging the data into a single Vector shapefile to create a layer of oil rig points within each SSM turtle buffer using ArcGIS Pro ver. 2.5 [113]. Drilling platforms were corrected by date to ensure they were in use during the date associated with the date of the 2 km turtle buffer.

Marine Cadastre, although very useful in acquiring data for the United States, is missing drilling platform data for other parts of our study area, specifically Mexico and Cuba. In order to understand if turtles that left the U.S. EEZ encountered oil threats, we used a world, oil exploration shapefile called PETRODATA, that covers oil drilling hotspots around the world [113]. Upon comparison with our existing dataset, 99.7% of oil rig points fall within the PETRODATA polygon for the United States, Gulf of Mexico, oil exploration polygon. Therefore, we felt it was comparable because no public oil rig data are available for Mexico at the time of writing this manuscript. No international turtle points fell within the confines of oil polygons so further calculations were not necessary.

### 2.6. Light Pollution Data

In 2011, the SUOMI VIIRS (Visible Infrared Imaging Radiometer Suite) satellite was launched to track multiple spatial data, such as snow and sea ice cover, active wildfires, sea and ice surface temperatures, and day/night light reflectance and radiance at high resolution [114]. We created our light pollution threat layer by combining all available light radiance raster data from NOAA's Earth Observation Group public download domain and averaging the total radiance for each pixel [114]. In total, 54 raster files, ranging from 13 January–8 March 2021, were combined by taking the average light radiance of each pixel, and then recording the average value of all pixels within each 2 km turtle buffer user R [105]. We assumed that the light radiance during the study period did not vary annually and that the light data we collected are representative of all years for which we have tracking data.

### 2.7. MPA, EEZ, and Proximity to Coast Data

We downloaded both the MPA Layer and EEZ Layer as vector layers from the Marine Cadastre data repository [111]. We created the coastal threat vector layer in ArcGIS Pro version 2.5 by making a 10 km buffer around all available land within the study region [112].

The SSM points that were >10 km from coastline, or within the U.S. EEZ or an MPA boundary were assigned a value of "0" to indicate threat absence, whereas points that were <10 km from the coast or outside of the U.S. EEZ or an MPA boundary were assigned a value of "1" to indicate threat presence.

As Marine Cadastre focuses on primarily U.S. waters, we needed data on international MPAs, specifically for Cuba and Mexico. Those data were downloaded from the IUCN's World Database on Protected Areas website [115]. We followed the same format of assigning values of "0" for turtles that were within the confines of those MPAs, and values of "1" for turtles outside the confines of those MPAs.

*2.8. Statistical Analyses*

To directly add and compare the effects of individual threats, we standardized all threat categories to a mean of zero and standard deviation of one, which allowed us to take the sum of all threats directly and create a cumulative threat score for each SSM point. Through preliminary data exploration, we discovered the data were substantially non-normal, and greatly spatially autocorrelated. As a result, we removed SSM points that were within 4 km of one another. Data removal reduced the number of SSM points from 8875 to 474. Despite removal, spatial autocorrelation still existed, but the degree to which it existed was reduced. Moran's I of cumulative impact scores changed from 0.458 ($p < 0.001$) at distance class I to Moran's I of 0.313 ($p < 0.001$). Complete removal of spatial autocorrelation from our dataset would have thinned the data to too few points to be able to run an analysis on; therefore, we decided to strike a balance between reducing the data, yet also minimizing the degree to which spatial autocorrelation existed in our dataset.

To test the prediction that individual or combined threats varied with species along a latitudinal gradient, we tested our thinned data using PERMANOVAs, a permutation-based test for significance as the data were still very non normal. PERMANOVAs were run using the R package 'vegan' [116]. We tested this using the interactive effects of species and latitude responding to threat. We included the threat x species interaction because we expected that species may vary in their response to spatially varying threats. Due to the potential for areas with variables of high threat to be clustered, data may not show a direct, linear relationship with latitude. To better understand any latitudinal relationships present in our data, we ran breakpoint regressions between latitude and threat (individual threats vs. latitude, and cumulative threat scores vs. latitude) to see if the breakpoint model was a better fit to the data. Due to the large spatial gap present and low sample size, we removed turtle 14, the lone hawksbill of this study, from this portion of the analysis. All data were given an alpha of 0.95 for detecting statistical significance. All data were analyzed using R [105].

In order to test the prediction that threats vary by species, we first calculated median values (due to the presence of outliers) of threat scores for each of the four numerical threats (Shipping = 0, Light = 1.3, Fishing = 1.2, Oil = 0) and used presence scores for the remaining three (Coast = 1, MPA = 1, EEZ = 1). Values above the median value for numerical threats or that had a score of 1 for categorical threats were categorized as "high threat," whereas values below the median or a score of zero, respectively, were considered "low threat." We then calculated the percentage of days during the study period an individual turtle encountered high threats by dividing the number of days a high threat was encountered by the sum of their SSM points. We calculated average values for each species from these percentages to understand how often sea turtle species were exposed to each threat during the study period. Preliminary data analysis discovered our data were very non-normal. Therefore, we ran PERMANOVAs on each threat category percent by species. Turtle 14, the single hawksbill captured for our study, was removed from this part of the analysis.

To test the prediction that turtles within MPA boundaries experienced lower threat, we recorded the mean time each individual turtle spent outside of an MPA using their SSM points. If individuals spent more than the mean value (23.6%) outside of an MPA, they were counted as a "non-MPA" turtle. Individuals that spent more time than the mean value

within an MPA were counted as "MPA" turtles. Three threat variables of continuous data (Light, Shipping, Fishing) that were categorized as high threat were compared between non-MPA and MPA turtles for statistical significance with a Welch's *t*-Test in R [104]. *t*-Test analyses were modified for unequal variance and if the equal variance assumption was violated.

To test the prediction that concentration of turtle SSM points is lowest in areas of high threat, we created a $10 \times 10$ km grid cell fishnet over the study region and then using the 'Spatial Join' tool to merge all points within each grid of the fishnet in ArcGIS Pro version 2.5 [110]. The number of turtle points within each grid cell was treated as a form of density for that specific area. Threat values of each grid cell were averaged if more than one turtle point was present. We then ran linear regression analyses to test for relationships between density and each individual threat, and several combinations of threat layers: All Combined Threats, Oil Threat (Oil, Shipping, and Coastal layers), Boat Threat (Fishing, Shipping, and Coastal, EEZ layers), and Fishing Threat (Fishing, Coastal, EEZ, and MPA layers). We additionally ran breakpoint regressions on our data to determine if density responded to threat nonlinearly or in linear segments using the R package 'segmented' [117].

To better understand threat interactions through time, we added the total number of high threat categories encountered on a given day for each SSM point and then plotted the threat on a color gradient (from 0–6) by month of the year and individual turtle, which clearly displays the daily number of high threats each turtle was exposed to during the tracking period. Turtles were sorted by capture location, MPA status, and species. Plotting data in such a manner can provide a visual interpretation of threats by individual, as well as show the timing of threats by month. All data were analyzed using R version 4.1.0 [104].

## 3. Results

Turtles were tracked for an average of 221.8 days [118]. The tracking period ranged from 16 days (minimum) to 1733 days (maximum). The mean number of days with SSM points varied by species. Mean tracking for loggerhead turtles was 363.1 days whereas green turtles and Kemp's ridleys were tracked for 185.7 and 89.5 days, respectively. Turtle #14, the only hawksbill turtle of this study, was tracked for 790 days (Figures 1 and 2) [118]. Seven of our turtles exhibited long-distance migrations between feeding and breeding grounds (Figure 3). Most notable was one loggerhead turtle that migrated from Key West, along the Florida Straits and up the east coast of Florida, and five green turtles, three of which migrated out of the Dry Tortugas National Park and Florida Keys National Marine Sanctuary to international waters, the Yucatan Peninsula, and the north coast of Cuba, well outside of the U.S. EEZ (Figures A1–A3 and Figures A5–A7). The fifth migratory green turtle, Turtle #16, made two migrations at similar times on two separate years. Increases in high threat exposure were found at similar times when migrations began for this individual (Figures 3 and A4).

Whether it be from shipping, fishing, oil rigs, light pollution, moving within coastal boundaries or outside MPAs, or leaving on long-distance migrations outside of the U.S. EEZ, most turtles experienced at least one threat on a daily basis. Of the total SSM points, 35.9% encountered no high threat exposure, 47% of which belong to just two individuals (turtle #25 and #4; Figure 3). However, only 19 points, representing five individual turtles, had a cumulative threat score of 0.

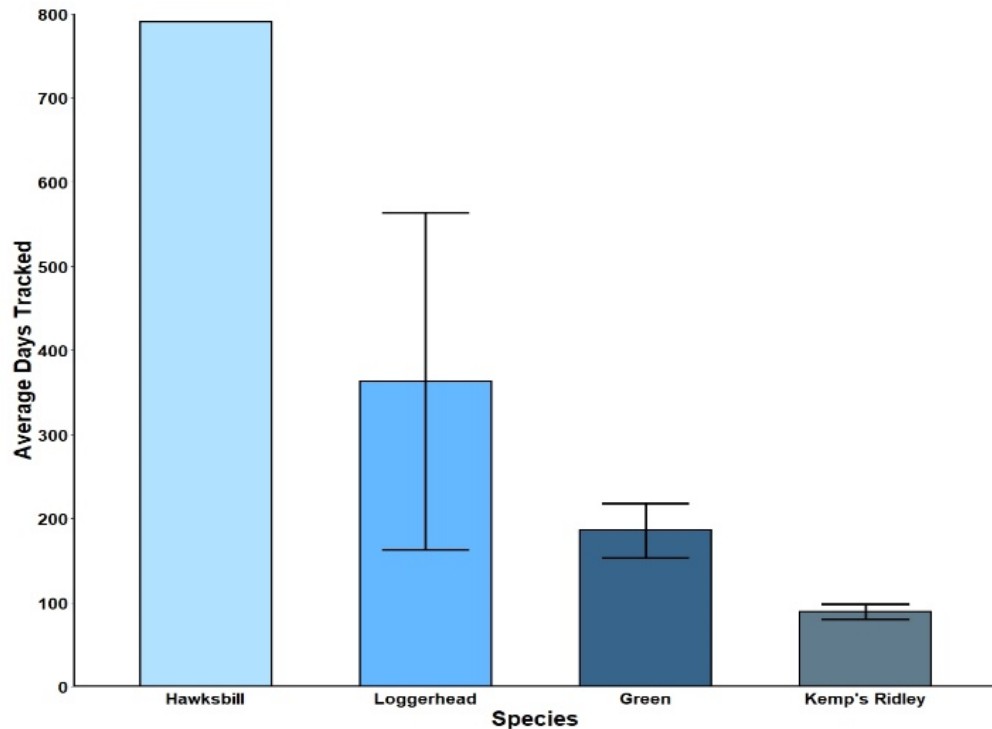

**Figure 1.** Average number of days tracked using mean number of state–space model (SSM) points per individual by species. Error bars represent standard error. Sample sizes are as follows: Hawksbill = 1, Loggerhead = 8, Green = 25, Kemp's Ridley = 6.

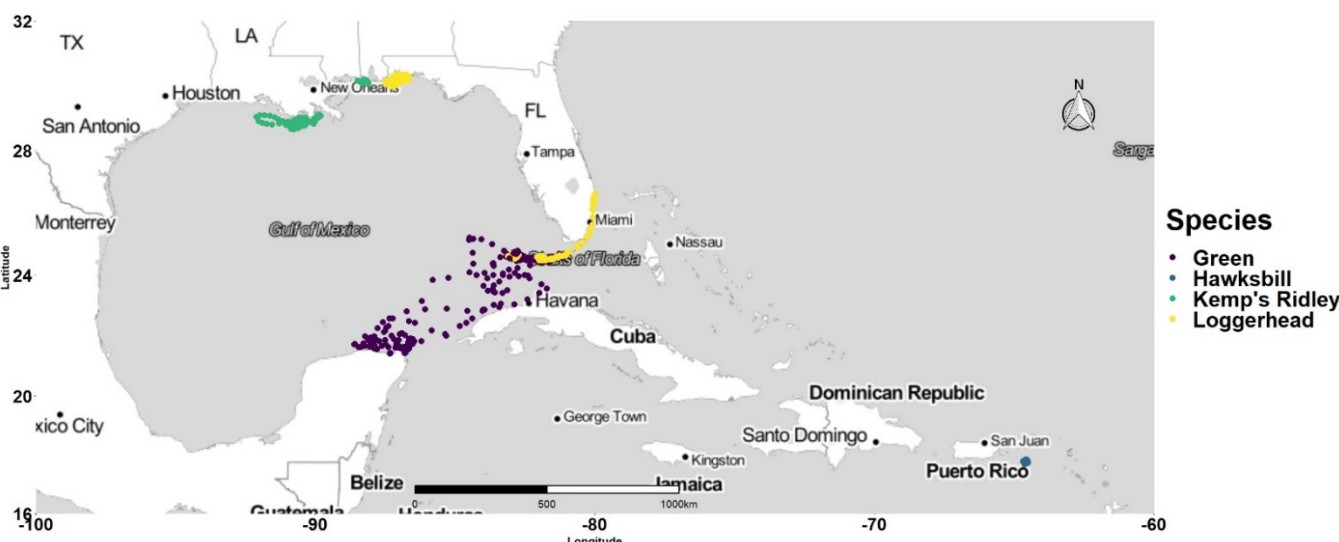

**Figure 2.** Map of study area showing state-space model (SSM) points by species. Turtle species are denoted by color.

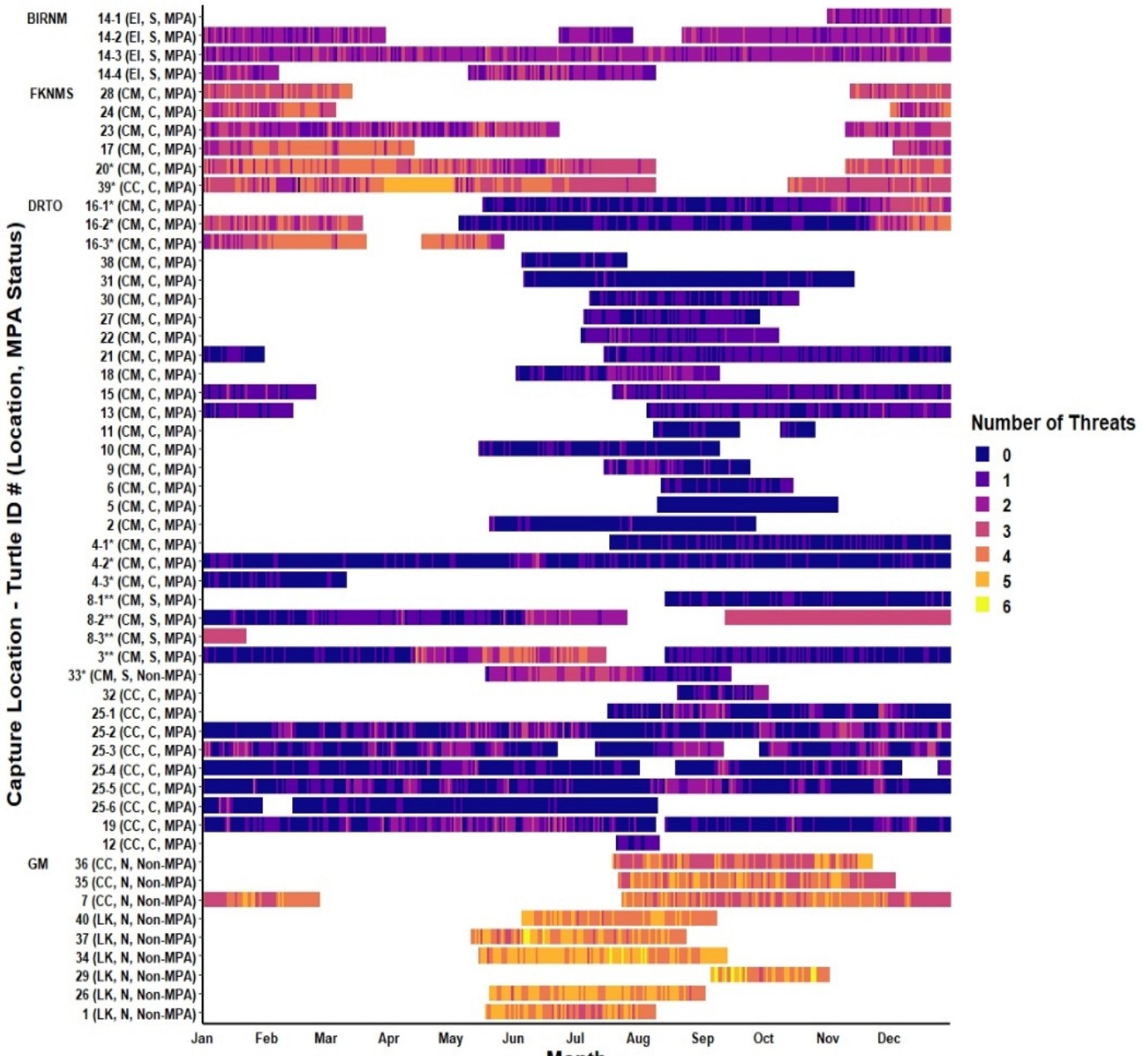

**Figure 3.** Number of high threat categories on a given day by each tagged turtle of our study. Data are sorted by capture location, species, geographic region (N = north, C = central, S = south), and Marine Protected Area (MPA) status (MPA or non-MPA designation). Missing data indicate lacking reliable tracking data or transmitters that terminated transmission before a given full calendar year. Turtles that have more than one calendar year of tracking data are displayed with a "-(number)" after their ID number. Asterisks (*) denote turtles that migrated, double asterisks (**) denote turtles that conducted migrations, but tracking data stopped before turtles returned to their foraging grounds. Acronyms are as follows: CC = loggerhead, CM = green, EI = hawksbill, LK = Kemp's ridley.

### 3.1. Latitudinal Gradient

Through our PERMANOVA analysis, we found a significant interaction between the effects of latitude and species on cumulative threat (total cumulative threats $F_{2,448} = 29.268$, $p = 0.001$; Figure 4), and five of seven individual threats (***Fishing*** $F_{2,448} = 9.499$, $p = 0.006$, ***Oil***— $F_{2,448} = 5.116$, $p = 0.012$, ***EEZ***—$F_{2,448} = 672.42$, $p = 0.001$, ***MPA*** $F_{2,448} = 77.430$, $p = 0.001$, ***Coastal*** $F_{2,448} = 10.236$, $p = 0.001$; Figure 5). Cumulative impact scores of threats showed higher values

in northern and southern latitudes of the study area, but were lowest at mid-latitudes within the Florida Keys National Marine Sanctuary and Dry Tortugas National Park (Figure 3). When the MPA threat was removed as a response, the cumulative values still exhibited a similar, significant trend ($F_{2,448} = 81.217$, $p = 0.001$), except that loggerhead turtles experienced higher threat in lower latitudes (Figure 6). Although the interaction between species and latitude was non-significant for Light Pollution and Shipping threats (***Light Pollution***—$F_{2,448} = 1.318$, $p = 0.253$, ***Shipping***—$F_{2,448} = 1.568$ $p = 0.177$), we did detect significant relationships with the individual effects of both light pollution (***latitude***—$F_{1,452} = 67.008$ $p = 0.001$, ***species***—$F_{2,451} = 21.142$ $p = 0.001$) and shipping (***species***—$F_{2,451} = 7.431$ $p = 0.001$). Breakpoint regression values for all threats combined and all individual threats were not significantly better fits than the original models.

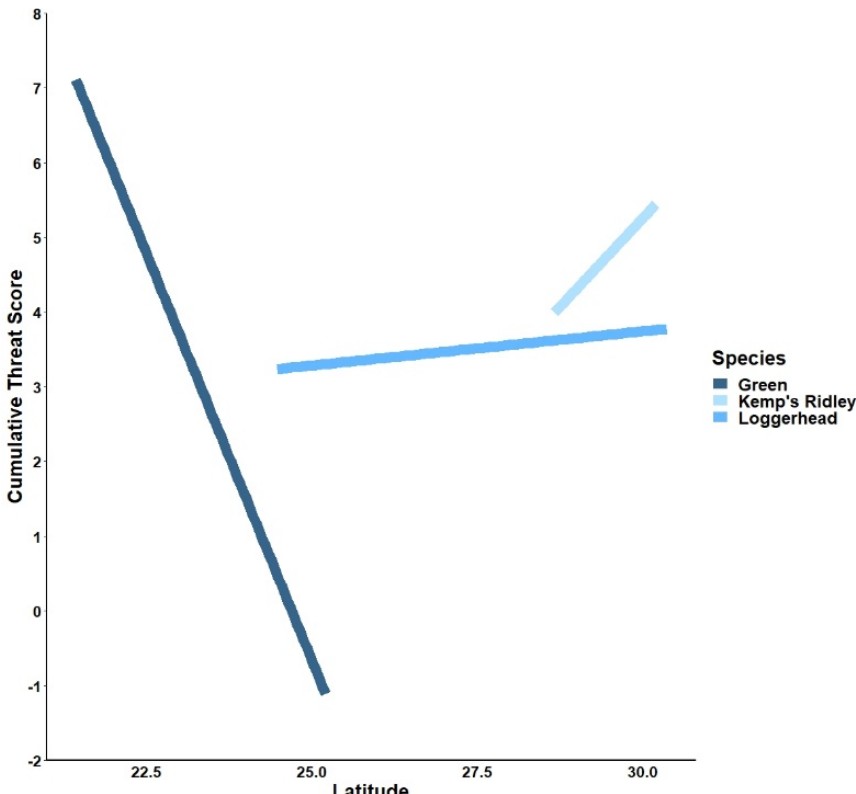

**Figure 4.** Interactive effects plot of Latitude × Species by the total score of all threats combined (cumulative impact score) using standardized scores. Standardizing threat scores reassigns values based on a mean of 0 and standard deviation of 1. Values are then re-calculated accordingly. PERMANOVA results detected a significant relationship between the interaction of species x latitude predicting threat ($F_{2,448} = 29.628$, $p = 0.001$). Low threat scores for green and loggerhead turtles coincide with the presence of the Florida Keys National Marine Sanctuary and Dry Tortugas National Park. Kemp's ridleys display lower threat scores in southern latitudes.

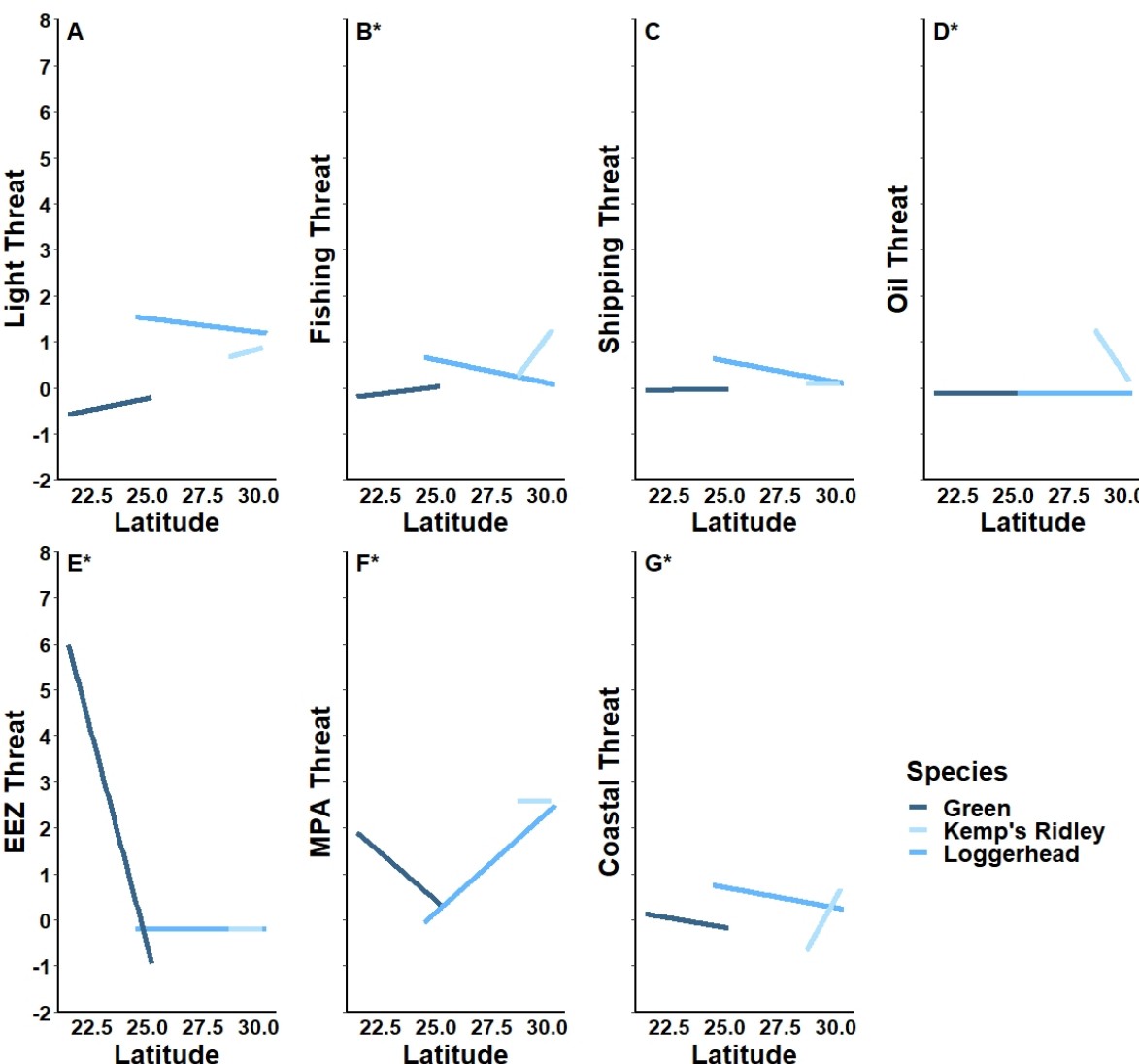

**Figure 5.** Interactive effects plot of latitude x species by the total score of all individual threats using standardized values. Standardizing threat scores reassigns values based on a mean of zero and standard deviation of one. Results from our PERMANOVA analysis found significant relationships between the interactive effects of latitude and species for five threats (*Fishing* (**B**) $F_{2,448} = 9.499$, $p = 0.006$, *Oil* (**D**)—$F_{2,448} = 5.116$, $p = 0.012$, Exclusive Economic Zone (*EEZ*) (**E**)—$F_{2,448} = 672.42$, $p = 0.001$, Marine Protected Area (*MPA*) (**F**) $F_{2,448} = 77.430$, $p = 0.001$, and *Coastal* (**G**) $F_{2,448} = 10.236$, $p = 0.001$). We found significant relationships on individual predictor variables and threat for both *Light Pollution* (**A**) (latitude—$F_{1,452} = 67.008$ $p = 0.001$, species—$F_{2,451} = 21.142$ $p = 0.001$) and *Shipping* (**C**) (species—$F_{2,451} = 7.431$ $p = 0.001$) threats. Asterisks (*) next to each embedded figure letter denote a significant result.

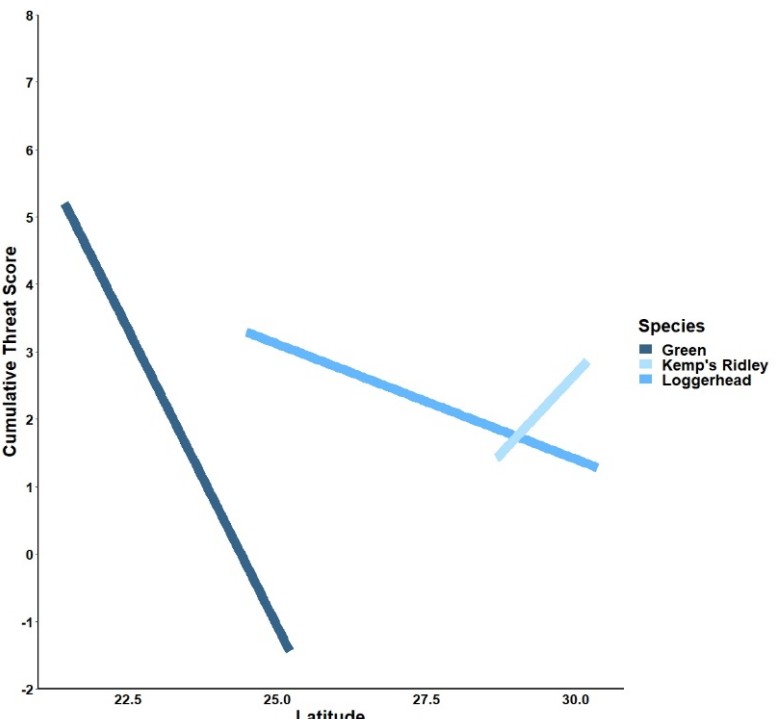

**Figure 6.** Interactive effects plot of latitude x species by the total score of all threats combined with Marine Protected Area (MPA) threat values removed (cumulative impact score) using standardized values. Standardizing threat scores reassigns values based on a mean of zero and standard deviation of one. PERMANOVA results show that removing MPA threat scores from our combined score still supports significantly lower threat values central to the Florida Keys National Marine Sanctuary and Dry Tortugas National Park for green turtles, but showed an increase in threat for loggerhead turtles ($F_{2,448} = 81.217$, $p = 0.001$). Kemp's ridleys also show lower cumulative threats in southern latitudes.

*3.2. Threat Exposure*

From our PERMANOVA analysis, we found that species varied in the degree to which they experienced five out of the seven threats (***Fishing***—$F_{2,36} = 11.03$, $p = 0.001$, ***Light Pollution***—$F_{2,36} = 7.9462$, $p = 0.002$, ***MPA***—$F_{2,36} = 52.837$, $p = 0.001$, ***Shipping***—$F_{2,36} = 34.523$, $p = 0.001$, ***Oil***—$F_{2,36} = 259.6$, $p = 0.001$; Figure 7). Kemp's ridleys were found to have the highest exposure of five threats (Fishing, Shipping, Light Pollution, Oil rigs and MPA threat). For three of these threats (Fishing Intensity, Light Pollution, and MPA threats), Kemp's ridleys encountered 100% high threat presence during their entire tracking period. They were also the only species tracked here to be found near oil rigs, with more than 50% of tracking days encountering high oil threat. Loggerhead turtles remained significantly nearer to the coast than other species, with more than 30% of tracking days found within 10 km of coastlines (coastal threat). Green turtles were the only species to move beyond the U.S. EEZ, with 5.2% of SSM points found in international waters. These points represent five individuals that migrated south, three to the Yucatan Peninsula and two to the northern coast of Cuba. Green turtles spent the most time within MPA boundaries and scored lowest in five out of the seven threat variables (Fishing Intensity, Light Pollution, MPA, Shipping, and Oil threats; Figure 7). Differences between species for distance to coast and EEZ threat exposure were found to be nonsignificant ($F_{2,36} = 0.6355$, $p = 0.504$ and $F_{2,36} = 0.7947$, $p = 0.479$, respectively; Figure 7).

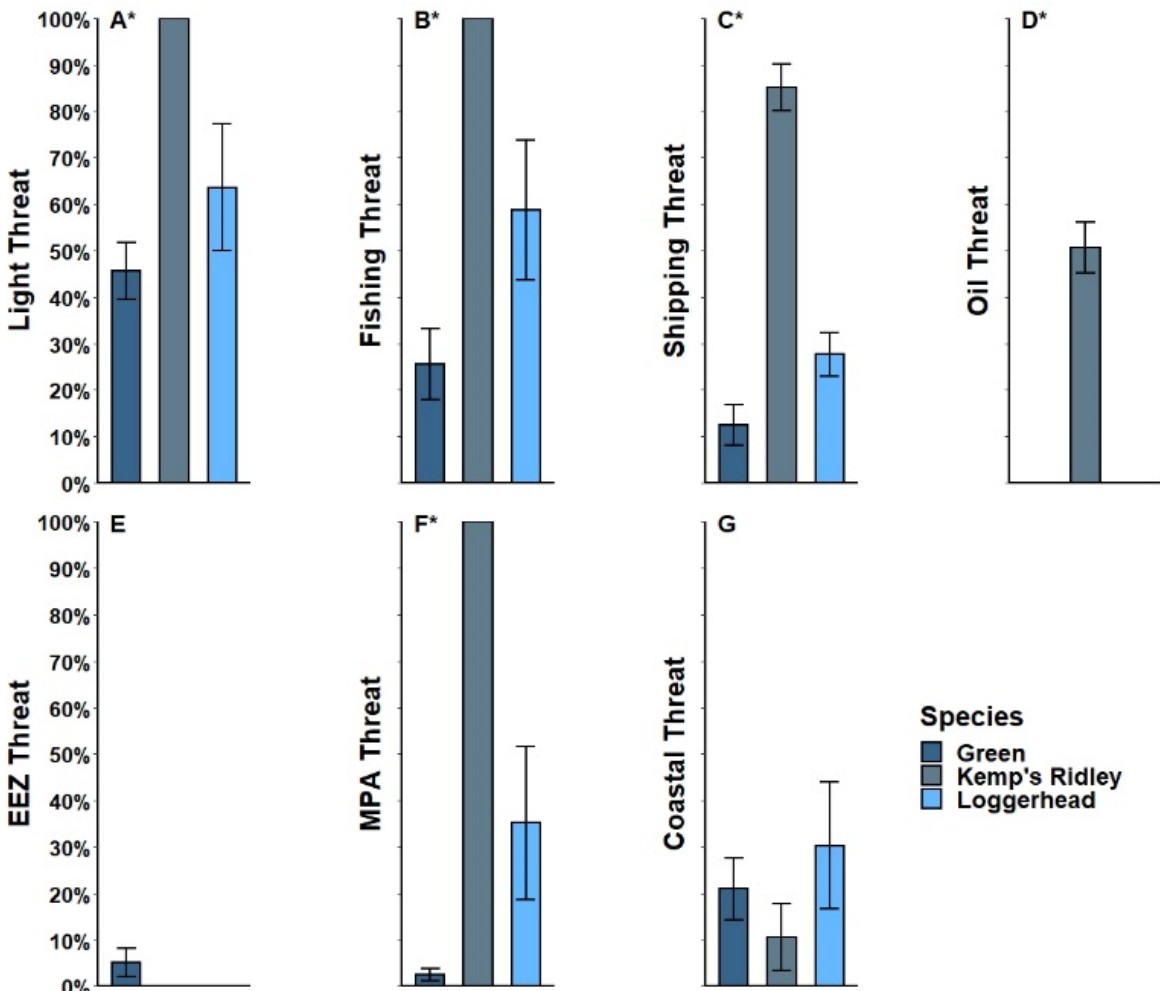

**Figure 7.** Average percentage of days a high threat was encountered by species. Error bars represent standard error. High threats were classified by threat scores above the median value for the given threat. PERMANOVA results found that threat scores significantly differed for 5 out of 7 threat variables (***Light Pollution*** (**A**)—$F_{2,36} = 7.9462$, $p = 0.002$, ***Fishing*** (**B**)—$F_{2,36} = 11.03$, $p = 0.001$, ***Shipping*** (**C**)—$F_{2,36} = 34.523$, $p = 0.001$, ***Oil*** (**D**)—$F_{2,36} = 259.6$, $p = 0.001$, and Marine Protected Area (***MPA***) (**F**)—$F_{2,36} = 52.837$, $p = 0.001$), for all of which Kemp's ridleys scored the highest. Nonsignificant variables were ***Coastal*** (**G**) and Exclusive Economic Zone (***EEZ***) (**E**) threats ($F_{2,36} = 0.6355$, $p = 0.504$ and $F_{2,36} = 0.7947$, $p = 0.479$, respectively). Asterisks (*) next to each embedded figure letter denote a significant result.

### 3.3. Threats and MPAs

From our analysis, we found a significant relationship between high threat exposure and the MPA status of turtles. MPA turtles had significantly lower frequencies of days where high threats were encountered compared with non-MPA turtles for all threat variables (***Fishing***—$T_{38} = -4.8428$, $p$ Please add the explanation for * in the figure. 0.001, ***Shipping***—$T_{38} = -4.4127$, $p \leq 0.001$, ***Light Pollution***—$T_{38} = -5.8619$, $p \leq 0.001$; Figure 8). High threat exposure for non-MPA turtles were over two-fold higher for Light Pollution (46.2% MPA vs. 94.1% non-MPA) and almost three-fold higher for Fishing (26.9% MPA vs. 90.0% non-MPA) threats. Additionally, Shipping threat exposure for non-MPA turtles was over four times higher than MPA turtles (14.3% MPA vs. 62.6% non-MPA; Figure 8).

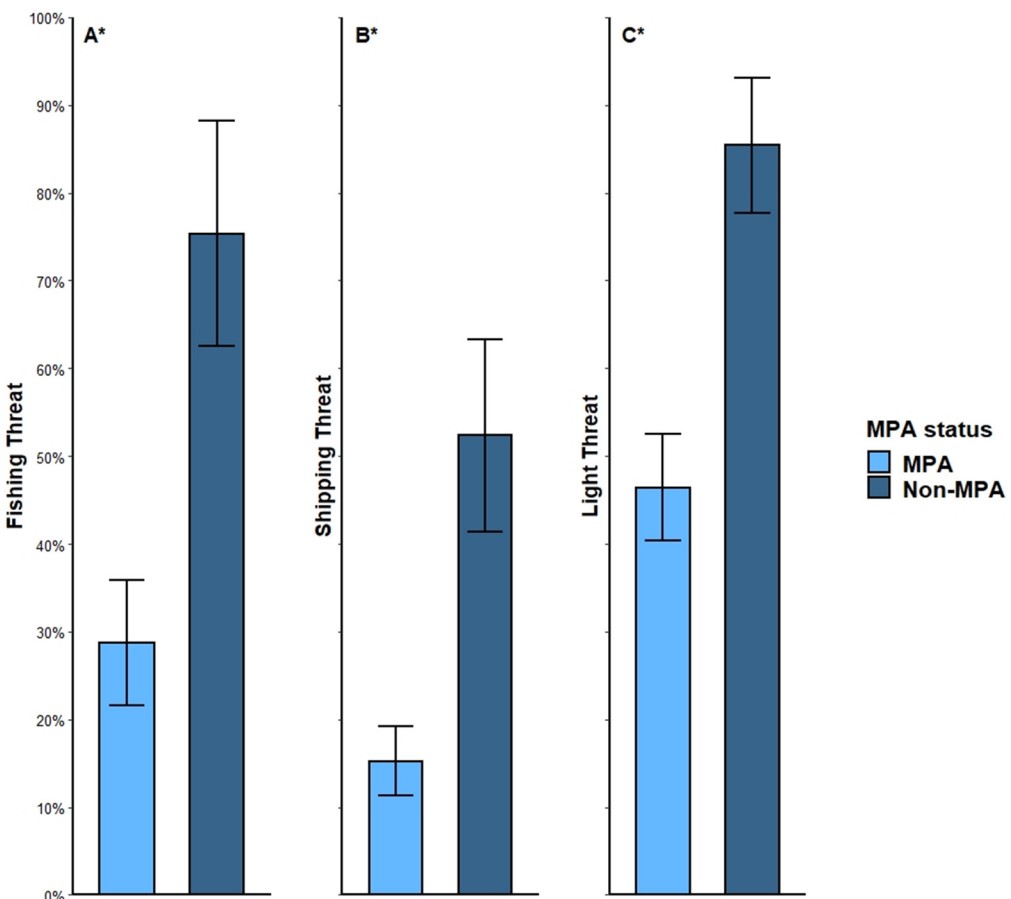

**Figure 8.** Average percentage of days a "high" threat of three categories (Fishing, Shipping, and Light Pollution) was encountered for "MPA" and "Non-MPA" turtles. "High" threats were classified by threat scores above the median value for the given threat. "MPA" and "non-MPA" classification were calculated by the mean percentage of time an individual turtle was in or out of an MPA boundary. "Non-MPA" turtles were significantly more likely to encounter a "high" threat than "MPA" individuals for all three tested threat variables (***Fishing*** (**A**)—$T_{38} = -3.399$, $p = 0.002$, ***Shipping*** (**B**)—$T_{38} = -3.191$, $p = 0.007$, ***Light Pollution*** (**C**)—$T_{38} = -3.7005$, $p = < 0.001$). Error bars represent standard error. Asterisks (*) next to each embedded figure letter denote a significant result.

*3.4. Concentration of Points and Threat*

No relationships were significant between concentration of SSM points for five of the individual threats (Shipping, Fishing, Light Pollution, Oil, Coastal, EEZ all $p > 0.05$). Additionally, Oil threat and Boat threat groups returned nonsignificant results from linear models. All combined threats had a significant effect on turtle concentration, ($F_{7,270} = 2.23$, $p = 0.0318$), and Fishing threat (Fishing, Coastal, EEZ, MPA layers; $F_{4,282} = 3.984$, $p = 0.004$), which was influenced by the low $p$-value from the MPA threat as a predictor to the response of concentration ($F_{1,276} = 12.53$, $p < 0.001$). However, due to the very low R-squared values (all threats combined = 0.030, Fishing Threat = 0.041, MPA = 0.039), these associations were not further investigated. To further test whether or not a relationship existed between MPA boundaries predicting turtle point concentration, a breakpoint regression was run on all individual variables and the cumulative threat, but was found to not be a significantly better fit than the initial, linear model.

## 4. Discussion

Male sea turtles in the Gulf of Mexico, the Caribbean, and Atlantic Coast of Florida are under pressure from many anthropogenic threats. Of the species we monitored, Kemp's ridleys had the highest threat exposure of all species for five out of seven threat categories—

three on a daily basis (100% of points in high exposure to Fishing, Light, and MPA threats). Kemp's ridleys were also the only species in this study to be present in areas with high oil rig threat presence. Loggerheads have been previously recorded in this location by previous studies [39,40,119], but not in our dataset. Loggerhead turtles consistently experienced the next highest level of threat, with the highest coastal exposure threat among all three species. Green turtles scored the lowest threat exposure among species for five of seven threats. However, they were the only species that left the U.S. EEZ and therefore scored highest in that category. The exposure of the aforementioned high threats varied significantly based on geographic location, MPA status, and species. Lastly, turtles that remained within MPA boundaries at least 76.4% of the time faced significantly less exposure to high threat compared to those outside of MPA boundaries for three of our numerical threats (Light Pollution, Shipping, and Fishing).

*4.1. Latitudinal Gradient*

We found significant differences between the interactive effects of species and latitude with all combined threats within our study site. Additionally, we found that the highest cumulative threat areas occurred in clusters north and south of the centrally located Florida Keys National Marine Sanctuary and Dry Tortugas National Park MPAs. Cumulative impact mapping using standardized values of all combined threats found a threat gradient, with relatively high threat in southern latitudes, which decreases around 24.5° N before increasing again in northern latitudes of the study area. When MPA threats are removed from our cumulative impact analysis, green turtles and Kemp's ridleys displayed similar results, but loggerhead turtles show a reverse trend of higher threat in southern latitudes and lower threat in northern latitudes. The area with lowest scores, whether MPA threats are included or not, coincided with areas represented by green turtles within the boundaries of two MPAs, the Florida Keys National Marine Sanctuary and Dry Tortugas National Park. We did, however, find an area of elevated threat within the MPA boundary, which is due to a major shipping lane that runs through the Florida Keys National Marine Sanctuary along the southern and western side of Key West to the rest of the Gulf of Mexico and the Florida Straits. Additionally, due to zoning and restricted take zones within the Florida Keys National Marine Sanctuary, the Dry Tortugas National Park had lower threat exposure than the Florida Keys National Marine Sanctuary.

Despite the cumulative trend of overall lower threat values within the confines of the Florida Keys National Marine Sanctuary and Dry Tortugas National Park, three individual threats (Light Pollution, Fishing, and Shipping) had higher scores in the reserve for both loggerhead and green turtles. Previous data have shown that migratory risks for female loggerhead turtles within this geographic area are high due to heavy boat traffic from fishing and ships with AIS transponders, particularly along the Florida Straits to the Atlantic coast of Florida [35]. Our data also show a similar trend for male sea turtles, with increasing threat values as turtles head northeast along the Florida Straits, and then increase again within the Gulf of Mexico. Despite an uptick in Light Pollution, Fishing, and Shipping threats within the Florida Keys National Marine Sanctuary along the Florida Straits, the lack of, or reduction in those and other threats within the Florida Keys National Marine Sanctuary and Dry Tortugas National Park compared to the Gulf of Mexico may indicate the effectiveness of MPAs within our study area.

Of all threats, fishing is often reported as one of the most impactful for turtles [35,36]. Previous studies within the Gulf of Mexico and Atlantic Ocean have found that fisheries bycatch represents a major source of mortality for sea turtles [35,40,41]. Hart et al. [40] also found that 77% of tracked turtles spent at least one day in high threat fishing locations. Our results also found that turtle points along the Atlantic Coast of Florida, the Florida Straits, and Gulf of Mexico had the highest fishing threat scores of all SSM points, which coincides with threats to female sea turtles [35].

Cumulative impact analysis, like the results in our study, has been used in other systems to develop ecosystem-based management practices [1,120,121]. Combined effects

of all threats can help shed insight as to where the areas under highest anthropogenic influence are located. Our study found the highest rates of cumulative impact within the Gulf of Mexico and Atlantic Coast, with the lowest rates of impact coinciding with the placement of the Florida Keys National Marine Sanctuary and Dry Tortugas National Park. By restricting human activities within large aggregate areas, threats to survival can be reduced.

*4.2. Threat Exposure*

We detected a significant relationship between individual threat exposure and species within our study. Overall, Kemp's ridley sea turtles faced the highest proportion threat exposure for five threats, with three high threats found at 100% of points (Shipping, Light Pollution, MPA) and the remaining two being 85.2% (Fishing) and 50.6% (Oil), respectively. Loggerhead turtles spent the most amount of time inside coastal waters (30.3%) and had the second highest scores of all tested threats for which they had values. Green turtles were the only species that left the U.S. EEZ (5.3%) but scored lowest for all remaining variables for which they had values except for coastal threats.

Several reasons are likely for Kemp's ridley sea turtles being exposed to the highest threats in our sample. For one, within our dataset, sites to which Kemp's ridley have high fidelity (aggregate clusters of SSM points) are unprotected, exposing turtles to increased threat exposure. Secondly, Kemp's ridley turtles are largely found within the Gulf of Mexico, which has the highest levels of fishing threat exposure from our study, as well as is documented as having the highest rates of recorded sea turtle bycatch within the United States [122]. Research has found that sea turtle bycatch in the United States from shrimp trawlers within the Gulf of Mexico was as high as 98% of total turtle bycatch from 1990–2007 [122]. Kemp's ridley turtles are also caught by recreational fishermen in the Gulf of Mexico. In one study, more than 12% of sampled turtles were found to have fishing hooks in their gastrointestinal tract [123]. Lastly, because they are a primarily Gulf of Mexico species, Kemp's ridleys were affected by the Deepwater Horizon oil spill in 2010, along with local populations of green, loggerhead, and hawksbill turtles [20]. Notable side effects from the oil spill were deformities in developing embryos, increased mortality, reduced immune systems, movement impairments, and other symptoms associated with oil toxicity [82,124–126].

Despite setbacks, Kemp's ridleys have been the subject of major efforts by multiple federal and non-profit agencies to recover viable populations. In the 20th century, Kemp's ridleys experienced a population collapse, losing more than 98% of their population between censuses in the 1940s and 1980s [44]. Despite a population rebound in recent decades, since 2010 nesting counts have plateaued in Texas and Rancho Nuevo, Tamaulipas, Mexico, where the majority of nests are found [127,128]. Part of this decline in nesting numbers has been hypothesized as a pulse event of a sudden drop in nesting females from the Deepwater Horizon oil spill [129]. Researchers have hypothesized that recent declines in nesting numbers are due to lack of available food resources. The lack of resources may stem from increases in neritic populations, competition with loggerhead turtles and fish species for food and discarded catch from fishing vessels, and increases in fishing pressure on crab and shrimp fisheries within the Gulf of Mexico [121]. Degradation of habitat could lead to further population declines in food resources for Gulf of Mexico resident sea turtles [127]. Due to the possibility that Kemp's ridleys receive food from fishing vessels and are conditioned to seek out humans for food, it is not unexpected that we found Shipping, Fishing, and Light Pollution threats to be the highest among the tracked males of this species.

The trend in stagnant nesting success of Kemp's ridleys could also be indicative of a mismatch in placement and functionality of MPAs for sea turtles within the Gulf of Mexico. Within the confines of our study, we found two clusters of high site fidelity for adult, male Kemp's ridleys off the coast of Louisiana (five males) and Alabama (being occupied by a single male), and one for male loggerhead turtles off of the Florida Panhandle. The three aggregates of SSM points almost entirely fall outside of any current, established MPA and

are subjected to heavy exposure to anthropogenic threats that could be detrimental to the success of the species as adult turtles represent the life stage contributing to future generational growth [130].

### 4.3. Threats and MPAs

At present data on the effectiveness of MPAs at protecting at-risk species are mixed. Although some studies have found positive effects in establishing MPAs for threatened species, others have indicated their effects are neutral or even negative. Without proper enforcement, anthropogenic activities such as fishing, can actually increase in MPAs, reducing their protective potential [110,131,132]. Revuelta et al. [131] found that without enforcement, MPAs in the Dominican Republic had significant increases in the aforementioned activities, putting resident animal populations within those MPAs at risk.

MPAs can also suffer from overpopulation of the species they intended to protect in the first place, which can lead to phenomena like overgrazing. Christianen et al. [133] found that MPAs within Indonesia were too small to manage the success of green turtle population increases, highlighting the need to incorporate necessary habitat into species management plans. Lack of suitable habitat further highlights that MPAs are not a universal solution to widespread habitat degradation within a region. Despite the shortcomings of MPAs around the world, and the controversy on the actual effectiveness of MPAs to protect marine life, our data indicate that MPAs significantly reduce exposure of three threats (shipping, light pollution, and fishing) in areas that overlap with MPA boundaries.

Effective MPAs have been found to limit both extractive (fishing, oil exploration, etc.) and non-extractive threats (recreational boating, shipping, etc.) [134] as well as include active law enforcement to curb illegal activity [135]. Our dataset further supports evidence that MPAs within the U.S. and Mexico are both effective and well-managed for male sea turtles. When placed in proper locations, and with adequate laws and enforcement, MPAs can help curb anthropogenic threats on sensitive marine species. Tracking data like those presented here have helped establish areas that reduce the severity of anthropogenic threats for resident animals [49,51,52]. Areas in Gabon, Mexico, and Indonesia with satellite tracking studies have contributed to the creation of MPAs, which have helped at-risk species, and many tracking papers like this one have found areas that MPAs could be created or expanded [42,53,136]. If properly placed, MPAs that encompass aggregate populations of male sea turtles could help reduce mortality from anthropogenic threats [136,137].

### 4.4. Future Research Directions

This paper currently contains the largest sample size of any male sea turtle study to date, and is the first paper to quantify threats for male sea turtles beyond the scope of fishing, recreation, or shipping by combining these threats with several other categories that have been shown to reduce sea turtle fitness. Although widely encompassing, some of our data layers fail to encompass seasonality, which could potentially increase threat exposure for turtles [138,139]. For example, every year in October, amateur boaters gather in large numbers to watch high-speed boat races through the Florida Keys [140]. This influx of high-speed boating could lead to an increase in threat exposure through boats that likely do not have AIS tags on board. Additional threats could also be seasonal, such as from tourism and subsequent pollution [139]. Further quantification of threats for male sea turtles in this area is warranted.

## 5. Conclusions

The data from this study provide us with more details on male sea turtle movements with emphasis on the Gulf of Mexico and the Caribbean, and the threats they are exposed to on a daily basis. Male sea turtles are an understudied group of their species, with most tracking studies focused on females due to being easier to tag when they nest. This study is the largest tracking study of male sea turtles to date, and reveals that although some males exhibit high site fidelity, others will conduct lengthy migrations that put them in

direct overlap with multiple anthropogenic threats. Using cumulative impact analysis, we found a latitudinal gradient, with higher threat scores in northern and southern latitudes, and an area of low threat for all species and two of three species when the MPA threat category is removed. Our data provide evidence to the effectiveness of MPAs when actively enforced. Lastly, we found that different species of male sea turtle face varying exposure of the same threat category, with Kemp's ridleys being the most widely exposed to high threat. Kemp's ridleys tracked here seem particularly vulnerable to the impacts from light pollution, fishing, and shipping, as exposure was very high compared to other species. Managers can use our results in the design of conservation measures for male sea turtles that may help curb the continued population declines of these sensitive, marine reptiles.

**Author Contributions:** Conceptualization, K.H. and J.I.W.; methodology, M.A., K.H. and J.I.W.; formal analysis, M.A., J.I.W. and K.H.; resources, K.H., M.A. and J.I.W.; data curation, K.H. and M.A.; writing—original draft preparation, M.A.; writing—review and editing, K.H. and J.I.W.; project administration, K.H.; funding acquisition, K.H., M.A. and J.I.W. All authors have read and agreed to the published version of the manuscript.

**Funding:** We acknowledge funding for various aspects of the tagging portion of this project from the U.S. Geological Survey (USGS) Ecosystems Wildlife program, the USGS Priority Ecosystems Science Program, the USGS Coastal and Marine Geology Program, the USGS Natural Resource Protection Program, and the National Park Service.

**Institutional Review Board Statement:** Numerous permits from several authorities across multiple states and territories have made our research possible. Permits issued to K. Hart include: MTP176; NMFS permits 20315, 17381, 13307, 22281; NPS permits DRTO-2018-SCI-0007, DRTO-2016-SCI-0008, DRTO-2014-SCI-0004, DRTO-2012-SCI-0008, DRTO-2010-SCI-0009, DRTO-2008-SCI-0008, BUIS-2016-SCI-0009, BUIS-2015-SCI-0012, BUIS-2014-SCI-0009, BUIS-2012-SCI-0002, BUIS-2011-SCI-0003, BISC-2019-SCI-0008,BISC-2018-SCI-0015; Federal Fish and Wildlife Territorial Permits STX036–11 and STX042–12 (St. Croix, U.S. Virgin Islands), and Louisiana Department of Wildlife and Fisheries Scientific Collecting Permits #WDP-19-006, LNHP-18-006, LNHP-17-001, LNHP-15-085. Sampling was approved under Institutional Animal Care and Use protocols USGS-SESC 2011-05, USGS SESC 2014-03, SER-BISC-BUIS-DRTO-EVER-Hart-Sea Turtles-Terrapins-2018-A2.

**Data Availability Statement:** Data collected for this study can be downloaded at https://doi.org/10.5066/P958OAKJ. See also Dryad Data Repository for code used to manipulate remotely sensed data at https://doi.org/10.5061/dryad.1rn8pk0ww.

**Acknowledgments:** We thank Kelsey Roberts for running the SSM, and many USGS employees and contractors for helping to capture male sea turtles in the water at various study sites. Research funding for M. Ashford was provided by the Eastern Shawnee Tribe of Oklahoma and the Watling Lab. Any use of trade, firm, or product names is for descriptive purposes only and does not imply endorsement by the U.S. Government.

**Conflicts of Interest:** The authors declare no conflict of interest. The funders had no role in the design of the study; in the collection, analyses, or interpretation of data; in the writing of the manuscript, or in the decision to publish the results.

## Appendix A

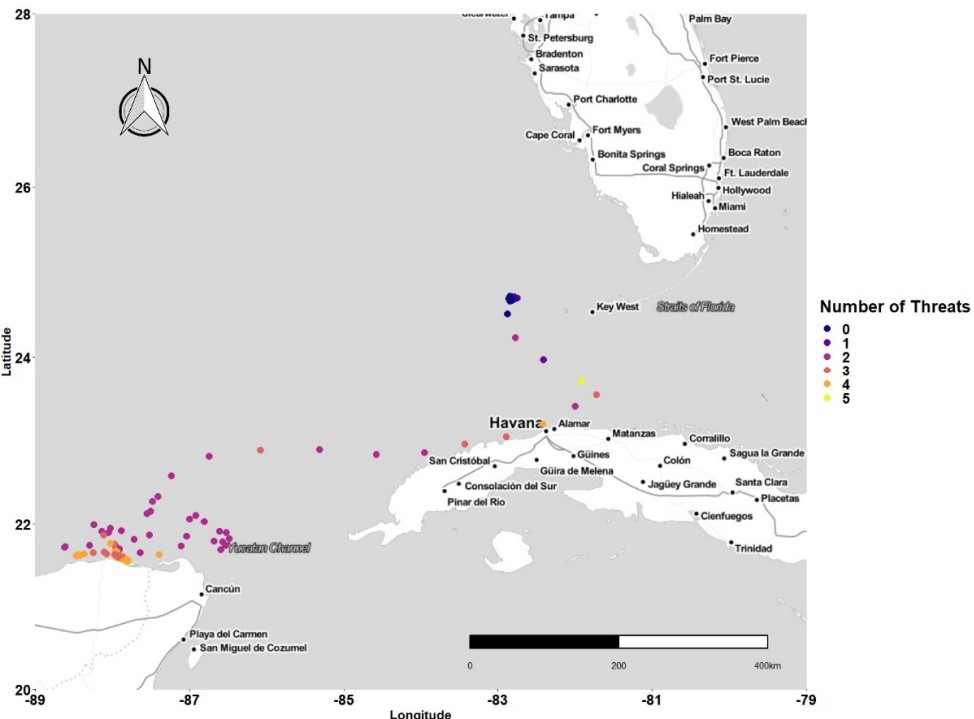

**Figure A1.** State-space model (SSM) points of Turtle #3, a green turtle (*Chelonia mydas*) that migrated from Dry Tortugas National Park to the Yucatan Peninsula between 13 April and 13 July 2019. Each color represents the number of unique high threat categories encountered on that given day within a 2 km buffer of the SSM point. Note that tracking data stops before this turtle returned.

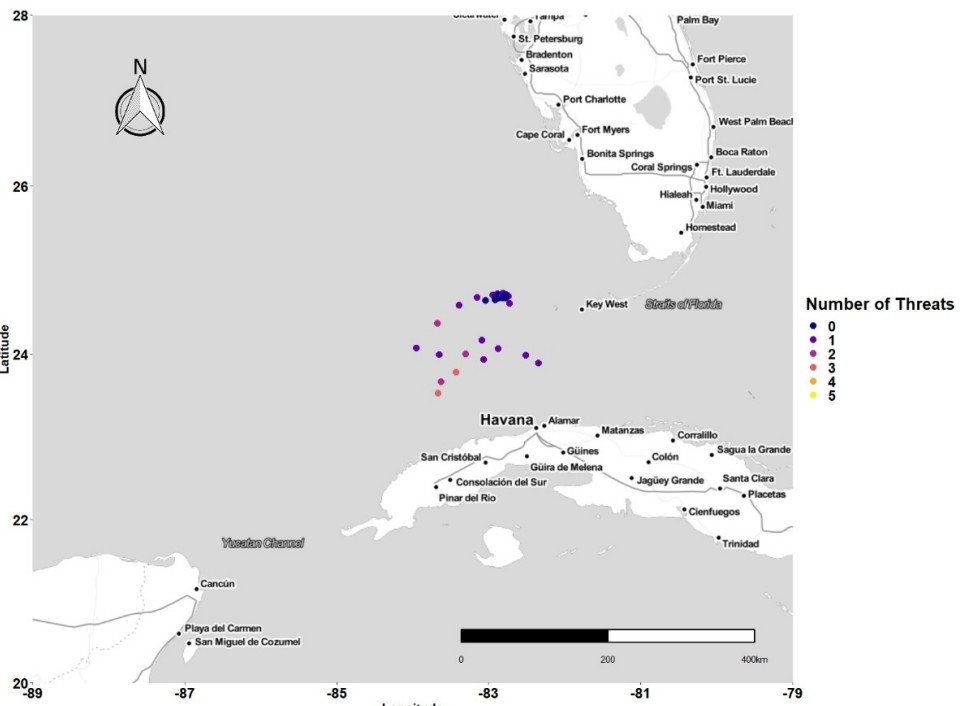

**Figure A2.** State-space model (SSM) points of Turtle #4, a green turtle (*Chelonia mydas*) that conducted migratory movements from and then promptly returned to Dry Tortugas National Park between 2 June and 18 June 2017. Each color represents the number of unique high threat categories encountered on that given day within a 2 km buffer of the SSM point.

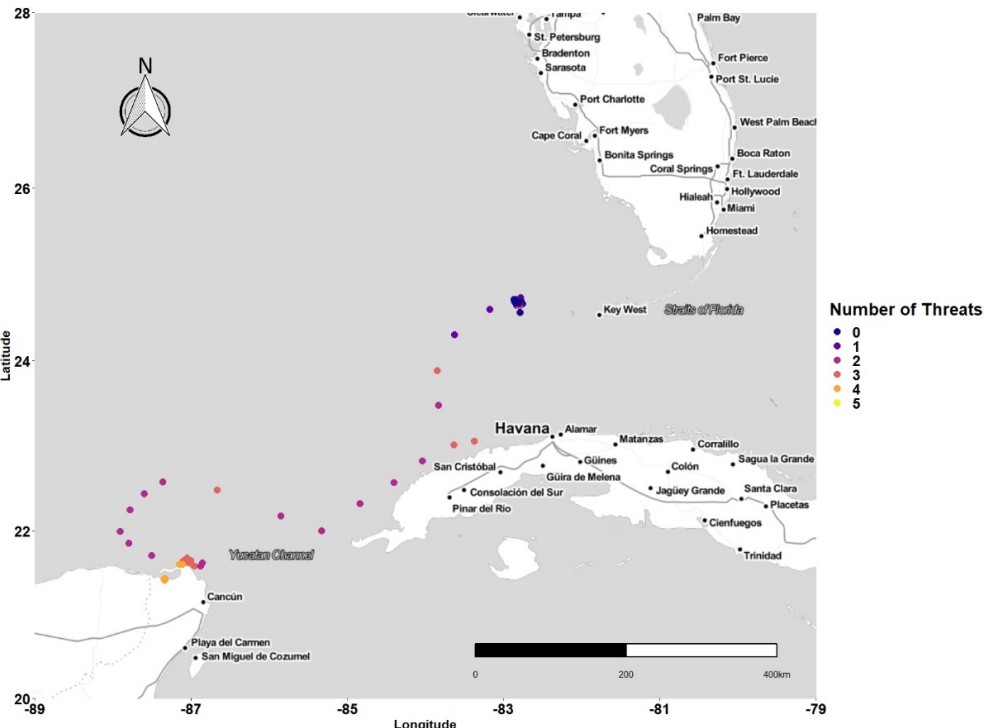

**Figure A3.** State-space model (SSM) points of Turtle #8, a green turtle (*Chelonia mydas*) that migrated from Dry Tortugas National Park to the Yucatan Peninsula between 7 June 2018 and 19 January 2019. Each color represents the number of unique high threat categories encountered on that given day within a 2 km buffer of the SSM point. Note that tracking data stops before this turtle returned.

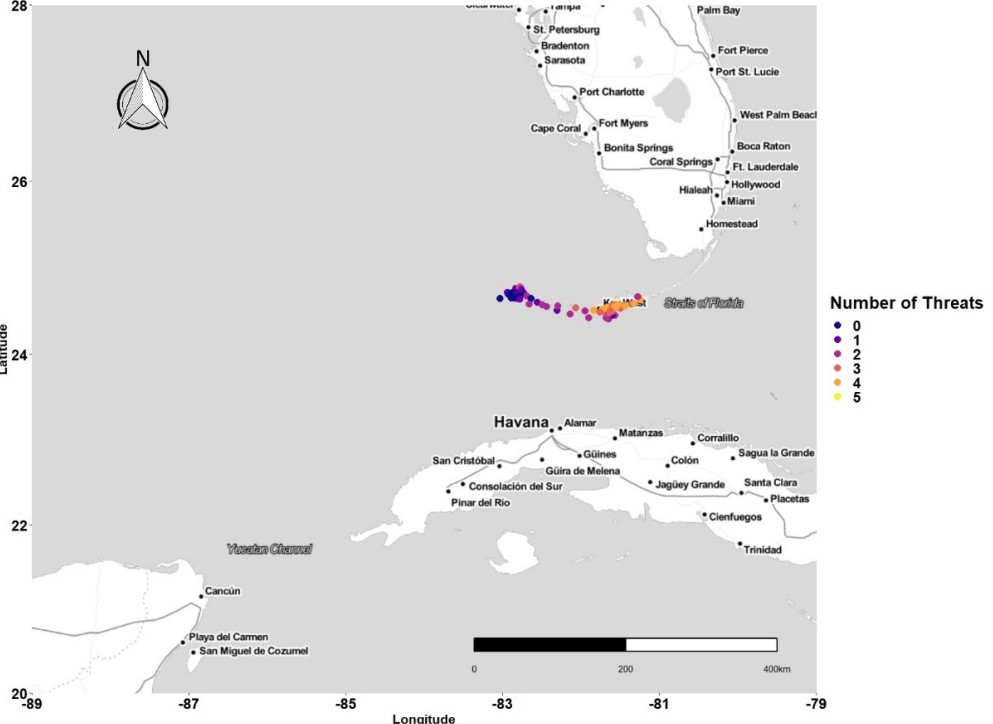

**Figure A4.** State-space model (SSM) points of Turtle #16, a green turtle (*Chelonia mydas*) that migrated from Dry Tortugas National Park to the Florida Keys National Marine Sanctuary along the Straits of Florida between late November and May 2016 and 2018. Each color represents the number of unique high threat categories encountered on that given day within a 2 km buffer of the SSM point.

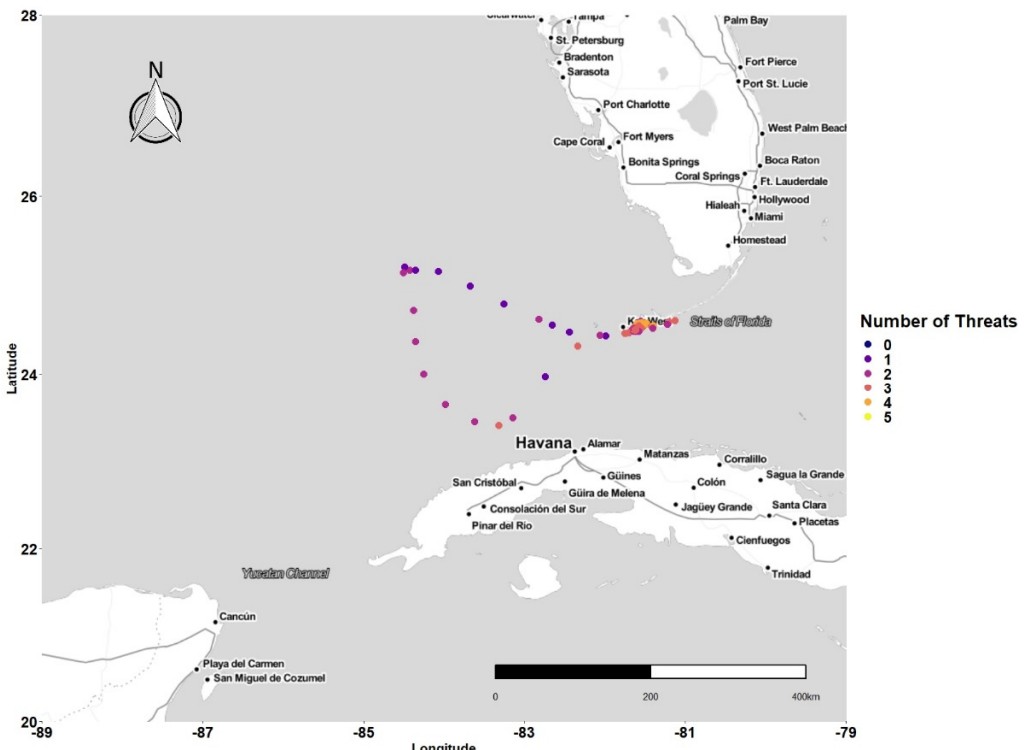

**Figure A5.** State-space model (SSM) points of Turtle #20, a green turtle (*Chelonia mydas*) that migrated from the Florida Keys National Marine Sanctuary near Key West towards the Gulf of Mexico and north coast of Cuba between 28 May and 19 June 2020. Each color represents the number of unique high threat categories encountered on that given day within a 2 km buffer of the SSM point.

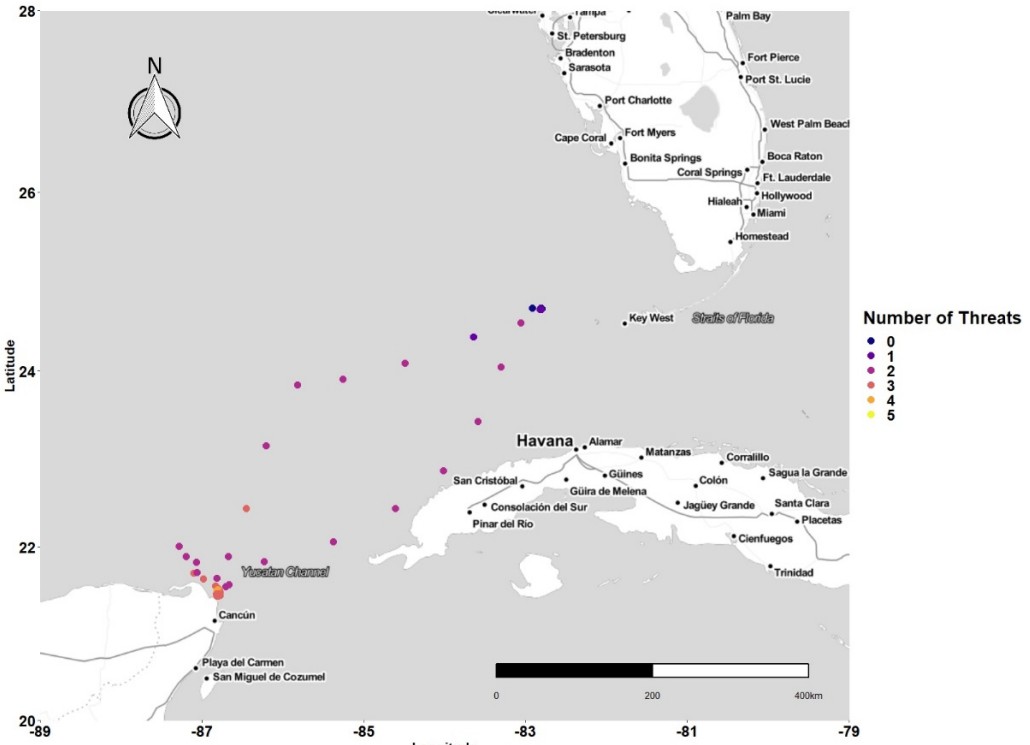

**Figure A6.** State-space model (SSM) points of Turtle #33, a green turtle (*Chelonia mydas*) that migrated from Dry Tortugas National Park to the Yucatan Peninsula and back between 22 May and 8 August 2015. Each color represents the number of unique high threat categories encountered on that given day within a 2 km buffer of the SSM point.

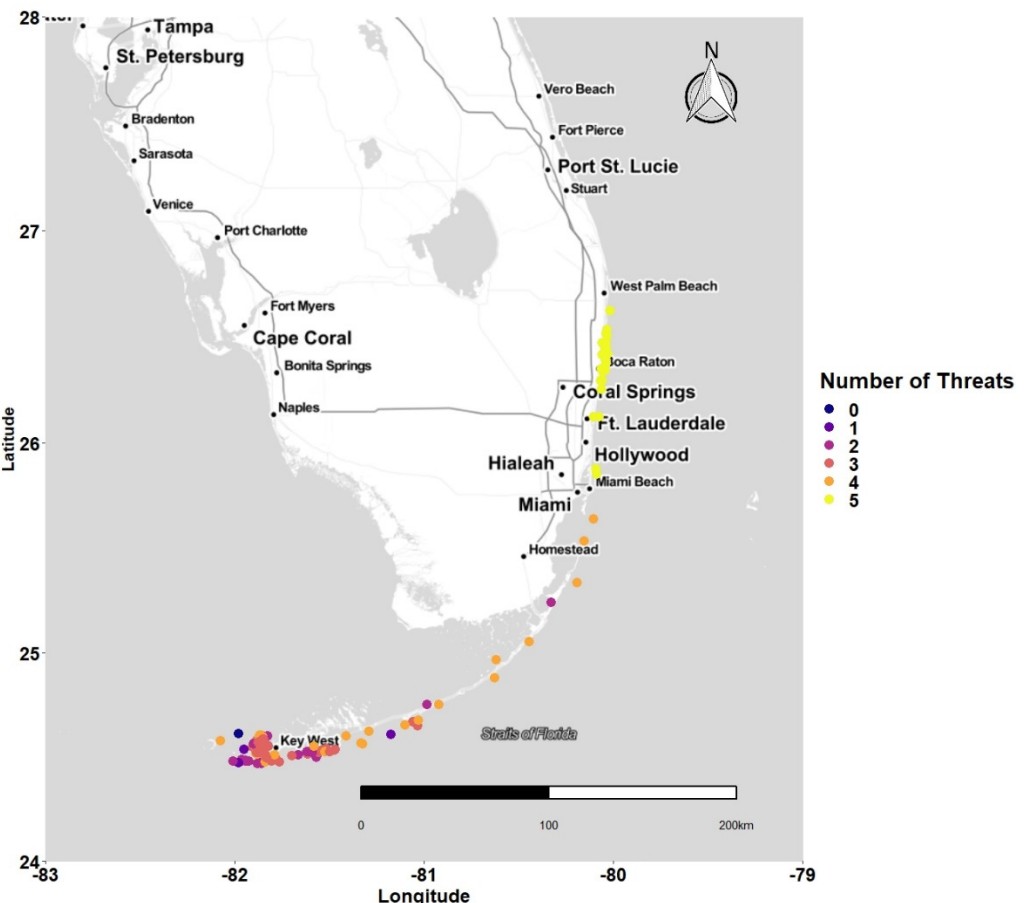

**Figure A7.** State-space model (SSM) points of Turtle #39, a loggerhead turtle (*Caretta caretta*) that migrated from the Florida Keys National Marine Sanctuary to the southeast Florida coast and then back via the Florida Straits between 5 March and 11 May 2020. Each color represents the number of unique high threat categories encountered on that given day within a 2 km buffer of the SSM point.

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
