# Peer review of "One Shell of a Problem: Cumulative Threat Analysis of Male Sea Turtles Indicates High Anthropogenic Threat for Migratory Individuals and Gulf of Mexico Residents"

_remotesensing, doi:10.3390/rs14163887_

Round 1

Reviewer 1 Report

Manuscript ID: remotesensing-1774914

One Shell of a Problem: Cumulative Threat Analysis of Male Sea Turtles Indicates High Anthropogenic Threat for Migratory Individuals and Gulf of Mexico Residents

Authors: Micah Nathaniel Ashford, James Ian Watling, Kristen Hart

Review

Really nice manuscript, just several small comments listed below, and deserves publication. I like additional materials presented.

However, knowing strict rules of publishing in some US organizations, I would start my review from some questions:

1.    Disclaimer: this will be removed after getting the permit? What authors supposed to do, if USGS will not issue this permit?

2.    Authors emails: for the third author, email did not match institution. Is this OK?

Comments:

Line 46: [6–10], not [6, 7, 8, 9, 10]

Line 49: [11–13]

Line 51: [14–16]

Line 58: [19,20]; further on – remove spaces in citations, see Template for authors

Line 61: [16,21], line 67 [23,34]

Line 69: [25–27]

Line 72: [23,26], line 74 [26,28]

Line 78: [27,35–37]

Line 81: 29–31,35,36,38.39–44]

Line 85: [45–48]

Line 100: [56–58].

I stop indicating changes to format of citations, in fact, all the rest require revision and changes.

Lines 106–112: please rewrite, avoiding nested parentheses; suggestion – split sentence inti several. Categories could be in normal font (not bold plus italics), as these are common.

Lines 196–199: there are punctuation problems in the sentence.

Lines 199–205: there are punctuation problems in the sentence. Check with Editor, if bulleted list could be used. Another comment on this – are these hypotheses tested?

Line 208: use styles, provided in the Template, for all subchapters; they should be numbered, like 2.1. Study Area and Species Collection (no : after the subtitles)

Lines 209–214: please split this part into smaller sentences, avoiding nested parentheses.

All R packages, e.g., ‘bsam’, ‘hDCRWS’ and further on, should be cited by original source, not only by citing research.

Line 209: 2009–2019, and please be consistent when using range, like in Line 262: 2015–2017, not 2015-2017

Line 276: 50–60%

Line 308: 2 km

Line 316: > 10 km; Line 318: < 10 km

Line 334: should this be p £ 0.0001?

Chapter “Statistical Analyses” lack references, perhaps there should be some, and in R, all used packages must be referenced.

Results

Line 394: 1733 days

Line 403: A1–A3, A5–A7

Lines 396 and 404: Turtle 14, but Turtle #16

Line 412 and for other figures: Figure 2. Not Figure 2:

Figure 3 could be enlarged, putting legend below the figure itself and using page width for graph.

Figure 3, Line 420: “Turtles that have more than one calendar year of tracking data are displayed with a “-#” after their:”… I failed to find this.

Lines 433–436: for the very different F p is given the same, p = 0.0001; is this true, or p < 0.0001?

Line 445: – (not - ) should be used

Figure 5. Caption, Lines 463–467, use different punctuation. Line 468: “* Denotes a significant result.” – I failed to find this at first, could be indicated better.

Line 476: certainly, with the same degrees of freedom, p < 0.0001.

Lines 481–483: at least, some p < 0.001

General comment for indicating F: could 2 decimals be used instead of 4? E.g., Line 505: “F2,36 = 0.6355, p = 0.504 and F2,36 = 0.7947” or “F2,36 = 0.64, p = 0.50 and F2,36 = 0.79,”

Line 512: what is T38?

Subchapters of Discussion should be numbered, 4.1., 4.2. and so on.

Line 621: 1990–2017

Line 637: spill [126].

I’d prefer to see “Future research directions” after the Conclusions, but this is just personal preference.

Back matter: use recommended format – “Conflicts of Interest: Declare conflicts of interest or state “The authors declare no conflict of in-terest.” Authors must identify and declare any personal circumstances or interest that may be perceived as inappropriately influencing the representation or interpretation of reported re-search results. Any role of the funders in the design of the study; in the collection, analyses or interpretation of data; in the writing of the manuscript, or in the decision to publish the results must be declared in this section. If there is no role, please state “The funders had no role in the design of the study; in the collection, analyses, or interpretation of data; in the writing of the manuscript, or in the decision to publish the results”.”

Supplements

This is Appendix, not Supplement, therefore labels Table A1, Figure B1 … B7, and must be referenced this way in the text.

References

Line 851: Žydelis is correct surname

Line 863: young-of-The-year  - I doubt for The

Line 884: Volume8, Issue24, December 2018, Pages 12656-12669

Line 1009: DOI: https://doi.org/10.3390/rs12091492 - use doi presentation consistently

Line 1017: 9, 1–14

Line 1022: is this DOI?

Line 1026: Sci Rep 6, 20625 (2016). https://doi.org/10.1038/srep20625

Line 1028: information missing

Line 1030: mistype Science (80-. ).

Line 1086: pages are 169–180

Author Response

Please see attached Word Document for responses to reviewer. Thank you so much for your time!

Reviewer 2 Report

This is a very interesting study containing a huge amount of information. It builds on the very limited existing information of higher exposure of male sea turtles to human threats, making it a key paper. Overall, it is generally well presented. I have just suggested some minor items to address, and look forward to seeing it published.

Line 117-118 (and Line 178, and discussion points where threats to males considered and potential impacts on populations), a key recent study that fits in here and stresses the higher threat and lower survival rates of adult male turtles versus female turtles due to anthropogenic activities in foraging habitat is the following, and should be drawn on here (it also strongly supports your findings regarding male turtles being at higher threat):

Schofield et al. 2020. Long-term photo-id and satellite tracking reveal sex-biased survival linked to movements in an endangered species. Ecology 101, e03027, doi: 10.1002/ECY.3027

Line 133 – I would avoid “outright flee” state instead there are trade offs. A good study to draw on here is this one:

Heithaus et al. 2007. State-dependent risk-taking by green sea turtles mediates top-down effects of tiger shark intimidation in a marine ecosystem. Journal of Animal Ecology. doi: 10.1111/j.1365-2656.2007.01260.x

Figure 3 – could this be simplified? i.e. this makes a useful supplementary figure. However, for a general reader it is very complex to interpret. I would present it as 2 graphs (1) % of time each turtle is exposed to threat and group it into bins, so number of turtles on y and % threat exposure on x; and (2) seasonality of threat, with the percentage of turtles in threat groupings per month. This way you could see if there is a particular month with greater threat quickly.

Note, the associated results does not clarify if any particular month has higher threat

Figure 8 – not sure why the y axis is exceptionally high/long on the page. Please amend.

Author Response

Please see attached Word document for replies to reviewer comments. Thank you so much for your time!
